# Novel Biomarkers and Imaging Indices for the “Vulnerable Patient” with Carotid Stenosis: A Single-Center Study

**DOI:** 10.3390/biom13091427

**Published:** 2023-09-21

**Authors:** Nikolaos Kadoglou, Konstantinos G. Moulakakis, George Mantas, Aris Spathis, Evangelia Gkougkoudi, Spyridon N. Mylonas, John Kakisis, Christos Liapis

**Affiliations:** 1Medical School, University of Cyprus, 2029 Nicosia, Cyprus; egkoug01@ucy.ac.cy; 2Vascular Surgery Department, Patras University Hospital, University of Patras, Rio, 265 04 Patra, Greece; konmoulakakis@yahoo.gr; 3Department of Vascular Surgery, Medical School, National and Kapodistrian University of Athens, Attikon University Hospital, 115 27 Athens, Greece; gmbest@windowslive.com (G.M.); kakisis@yahoo.gr (J.K.); liapis@med.uoa.gr (C.L.); 42nd Department of Pathology, National and Kapodistrian University of Athens, 115 27 Athens, Greece; arisspa@gmail.com; 5Department of Vascular and Endovascular Surgery, University Hospital of Cologne, 50937 Cologne, Germany; spyros.mylonas@gmail.com; 6Department of Vascular and Endovascular Surgery, Athens Medical Center, 106 73 Athens, Greece

**Keywords:** carotid atherosclerosis, vulnerable patient, matrix metalloproteinases, cardio-ankle vascular index, carotid plaque instability

## Abstract

Background: We investigated the relationship of matrix metalloproteinases (MMPs), cardio-ankle vascular index (CAVI), and Gray-Scale Median (GSM) score with the severity and vulnerability of carotid atherosclerosis and major adverse cardiovascular events (MACE) during follow-up of carotid artery revascularization. Methods: We enrolled 262 patients undergoing carotid revascularization therapy (GRT), 109 asymptomatic patients with low-grade carotid stenosis (40–70%) receiving conservative treatment (GCT), and 92 age- and sex-matched control subjects without carotid atherosclerosis (GCO). All participants underwent carotid ultrasound and we assessed at baseline clinical parameters, metabolic profile, CAVI, GSM, and circulating levels of hsCRP, MMP-3,-7,-9, and TIMP-1. Results: Both GRT and GCT presented with elevated CAVI, MMPs, and TIMP-1 levels compared to GCO (*p* < 0.001). The escalation highly correlated to the presence of symptoms or paralleled the degree of carotid stenosis (*p* < 0.001). During follow-up (mean duration: 55 months), 51 GRT patients experienced MACE unrelated to the revascularization procedure. Within GRT, diabetes (HR: 2.07; CI: 1.55–2.78, *p* < 0.001), smoking (HR: 1.67; CI: 1.35–1.95, *p* < 0.001), high CAVI (HR: 1.22; CI: 1.09–1.43, *p* = 0.023) and MMP-9 (HR: 1.44; CI: 1.29–2.15, *p* = 0.005), and low GSM (HR: 1.40; CI: 1.16–2.12, *p* = 0.002) independently predicted MACE occurrences, despite the optimum medical therapy. Conclusions: Novel imaging and biochemical biomarkers were positively associated with atherosclerosis severity, while CAVI, MMP-9, and low GSM showed a positive, independent relationship with MACE after carotid revascularization, describing “vulnerable patients”.

## 1. Introduction

Cerebrovascular ischemia is among the most important causes of cardiovascular morbidity and mortality in developed countries, and one of the leading causes of social isolation with a remarkable socio-economic burden on the healthcare system [1]. The presence of severe carotid atherosclerosis reflects the high atherosclerotic magnitude and cardiovascular risk of those patients [2]. Current guidelines have reported significant carotid atherosclerosis as a major therapeutic target of primary and secondary prevention, including intensive cardiovascular risk factors modification, such as lipid lowering [3]. The amount of carotid stenosis has been previously associated with elevated risk for ipsilateral ischemic strokes or transient ischemic attacks (TIAs) and determines the timing of carotid revascularization [4]. In addition to the degree of carotid stenosis, the vulnerable texture of the carotid plaques, assessed by imaging modalities and biomarkers, poses prognostic power for carotid atherosclerosis evolution [5]. A growing body of evidence has supported the clinical importance of “vulnerable plaques” characterized by instability and high propensity to rupture leading to cerebrovascular events [6,7]. Nowadays, it is believed that acute plaque destabilization is not entirely a local event, but it reflects a generalized high risk for acute cardiovascular events [8]. Many researchers have used the term “vulnerable patient”, defined as a patient with established atherosclerosis throughout the artery tree, who clusters a number of cardiovascular risk factors, traditional and novel, further precipitating plaque vulnerability and exaggerating the overall cardiovascular risk [9,10]. The presence of calcification at any arterial district may predict atherosclerosis [11]. More frequent major adverse cardiovascular events (MACE) than others with manifestations of atherosclerosis characterize “vulnerable patients”.

There are many imaging indices of carotid plaque vulnerability. Gray-Scale Median (GSM) score, a valid index of carotid plaque echogenicity, considerably relates to plaque [12] and patients’ vulnerability [13]. It is feasible, highly available, and reproducible [14]. Most recently, the cardio-ankle vascular index (CAVI), which is independent of blood pressure, has been developed as a surrogate index of arterial stiffness. It is associated with the presence of a composite of coronary and carotid atherosclerosis [15] and cardiovascular risk in several populations, including patients with carotid atherosclerosis [16,17,18,19].

As an adjunct to vascular imaging, a long list of circulating molecules, known as biomarkers, has been proposed for the detection of vulnerable plaques and “vulnerable patients” [20,21]. Matrix metalloproteinases (MMPs) are a well-established family of protases unambiguously contributing to atherosclerotic plaque progression and regression [22,23]. Previous studies have demonstrated the association between high levels of MMPs, such as MMP-3, MMP-7, and MMP-9, and low levels of their inhibitor (tissue inhibitor of matrix metalloproteinases-1, TIMP-1) within either carotid plaque specimens or blood samples from patients with vulnerable carotid plaques [24,25]. Notably, MMPs have shown considerable association with acute coronary syndromes [26], while their inhibitor (TIMP-1) has shown a protective impact against atherosclerotic plaque destabilization [27]. Furthermore, the MMP-9/TIMP-1 ratio has been proposed as a stronger independent predictor of coronary and carotid atherosclerosis [28,29]. Despite the plethora of data, circulating levels of MMPs or TIMP-1 have still not been widely accepted for “vulnerable patients” detection.

The aim of the present study was to investigate the relationship of CAVI and novel biomarkers (such as MMPs and TIMP-1) with: (1) the presence and the ultrasonographically quantified severity of established carotid atherosclerosis; (2) the carotid plaque vulnerability; (3) the long-term incidence of MACE after carotid revascularization.

## 2. Materials and Methods

### 2.1. Study Groups

This is a prospective, observational study of consecutive patients with established extracranial carotid atherosclerosis of the unilateral or bilateral internal carotid arteries (ICA). A total of 402 patients (Department of Vascular Surgery, Attikon Hospital, National Kapodistrian University of Athens) with extracranial ICA stenosis, diagnosed from 2010 to 2016, were prospectively analyzed and included in this study. Patients with common carotid artery (CCA) occlusion were excluded.

The complete medical history for each patient was extracted and reviewed, including atherosclerotic clinical symptoms, risk factors such as age, gender, presence of coronary artery disease, diabetes mellitus, hyperlipidemia, hypertension, smoking, and previous neurologic events (asymptomatic, stroke, transient ischemic attack (TIA), and amaurosis fugax).

Those patients were divided into the following groups:Group of revascularization therapy of carotid stenosis (GRT, N = 288 patients): Patients fulfilling the criteria for carotid revascularization with either carotid endarterectomy or carotid artery stenting (CAS) [5]. This group was further subdivided into patients with symptomatic carotid artery stenosis ≥ 50% (n = 104), or asymptomatic carotid artery stenosis ≥ 70% (n = 184). The symptomatic subgroup had presented within the last 6 months with neurological symptoms and/or brain scan findings compatible with ipsilateral to the carotid stenosis transient ischemic attack (TIA), stroke, or amaurosis fugax. Asymptomatic patients were free of related neurological symptoms. In order to limit the possibility of patient misclassification, all patients had undergone brain computed tomography (CT) scan or brain magnetic resonance imaging (MRI) examination, when CT findings were questionable cerebral infarcts. The degree of carotid stenosis was calculated from carotid ultrasound or magnetic angiography of both carotids.Group of conservative treatment of carotid stenosis (GCT, N = 114): Asymptomatic patients with significant carotid atherosclerotic plaques causing stenosis between 40% and 70%. All patients were evaluated by a vascular surgeon and neurologist and underwent at baseline anthropometrical/clinical assessment, carotid ultrasound, blood analyses, brain computed tomography (CT) scan, and/or brain magnetic resonance imaging (MRI), when CT findings were questionable.Group of controls (GCO group, N = 92): Age- and sex-matched individuals served as controls and were only assessed at baseline. Those subjects were selected after testing for exclusion criteria. All of them had visited our hospital for regular health check-ups. They had a maximum of two classical cardiovascular risk factors (such as hypertension, diabetes, dyslipidemia, smoking, or family history of premature coronary artery disease), but no evidence of overt cardiovascular disease. Moreover, their carotid and lower limb arteries were free from atherosclerotic plaques based on vascular ultrasound and they had a functional test negative for myocardial ischemia within the last year.

All patients with carotid stenosis were under intensive statin treatment (LDL target < 70 mg/dL) and acetyl salicylic acid (100 mg o.d.), unless it was contra-indicated. General exclusion criteria were the following: life-threatening diseases (e.g., malignancies), significant peripheral arterial disease (ankle-brachial index (ABI) < 0.9), atrial fibrillation, severe liver (AST ≥ 3 times the upper normal limit) or renal impairment (creatinine ≥ 2 mg/dL), heart failure classification stage II–IV according to the New York Heart Association (NYHA), chronic inflammatory or autoimmune diseases, as well as acute infection at study entrance or recent major event (within the last month) such as major trauma, surgery, or cardiovascular ischemic attack. Patients with peri-procedural complications (TIAs or stroke) were excluded as well from the analysis.

Written informed consent was obtained from all participants before enrollment and all procedures were performed according to the principles of the Helsinki Declaration and were approved by the hospital’s human ethics committee. 

### 2.2. Study Design

The abovementioned assessment was conducted in all participants at baseline. Then, patients with carotid atherosclerosis (groups GCT and GRT) were followed-up for at least 24 months. GC served as controls and we obtained data only at baseline. Physicians were asked to optimize the guideline-directed medical therapy of all patients and to encourage them to incorporate a healthier lifestyle. After the revascularization procedure, GRT remained on dual antiplatelet therapy (clopidogrel 75 mg plus acetyl salicylic acid 100 mg o.d.) for 6 months and then on life-long clopidogrel therapy, unless it was deemed medically necessary to select another antiplatelet agent. During follow-up, we recorded deaths or the occurrence of any MACE: acute coronary syndromes (myocardial infarction or not), coronary revascularization, TIA, ischemic stroke, carotid artery revascularization, and carotid artery restenosis requiring re-revascularization (either endovascular or open surgery), based on medical records. Patients with angiographic stenosis ≥ 50% found either in stents or at lesion sites of previously carotid endarterectomy underwent re-revascularization. A multi-disciplinary team (vascular surgeons and cardiologists) peer-reviewed all medical records and the decision of fulfilling or not MACE criteria was taken. Patients with at least one MACE were assigned to the MACE subgroup and the rest to the non-MACE subgroup for analysis purposes.

### 2.3. Clinical Data Collection

Body-mass index (BMI) and waist-to-hip ratio (WHR) were obtained in all participants at the study entrance. Blood pressure was measured twice, after keeping participants at a sitting position for 15 min. There was a 5 min interval between the two measurements and the mean value was estimated. Based on a structured questionnaire, we recorded all medications and co-morbidities, defined as follows: hypertension: BP ≥ 140/90 mmHg measured on repeated occasions or antihypertensive therapy; hyperlipidemia: fasting serum low-density lipoprotein cholesterol (LDL-C) ≥ 160 mg/dL or statin therapy; active smokers: current or within the previous 6 months; diabetes mellitus: HbA1c ≥ 6%, or antidiabetic drugs; coronary artery disease (CAD): history of stable or unstable angina, myocardial infarction, percutaneous or surgical coronary revascularization. In our study cohort, all patients with carotid stenosis were already on statin therapy and aspirin as preventive measures. CAVI (Vasera VS-1500, Fukuda Denshi, Tokyo, Japan) was calculated in all patients along with ABI. 

### 2.4. Carotid Ultrasound Examination

All patients underwent carotid ultrasound examinations at baseline (whole study cohort) by two experienced operators. Based on a validated protocol, we graded the percentage of arterial stenosis by measuring the Peak Systolic Velocity (PSV) and internal carotid artery (ICA)/common carotid artery (CCA) PSV ratio [30]. The analysis of morphological and textural features of carotid plaques on ultrasound images was conducted offline and the process has already been described elsewhere by our group [31]. 

The Gray-Scale Median (GSM) score is a valid, quantified index of carotid plaque echogenicity, inversely associated with carotid plaque vulnerability [32]. In asymptomatic patients, we averaged GSM scores from all carotid plaques, while in symptomatic patients we reported the GSM score of the culprit lesion, ipsilateral to brain infarct. GSM calculation has been validated in our laboratory, where it is used in routine clinical practice. 

### 2.5. Blood Assays

For all patients, blood samples were obtained at baseline after an overnight fast, between 8.00 and 10.00 a.m. FPG and lipid parameters were all measured in an automatic enzymatic analyzer (Olympus AU560, Hamburg, Germany). The following MMPs were measured: MMP-3, MMP-7, MMP-9, and its inhibitor TIMP-1. Each blood sample was centrifuged, and their serum was frozen (storing temperature −74 °C). The Luminex xMAP technique was used to measure MMPs and TIMP-1 in serum samples. xMAP technique is a widely used technique for measuring multiple biomarkers and proteins using color-coded microsphere beads and biotinylation antibodies. High-sensitivity C-reactive protein (hsCRP) was assayed using an immunoturbidimetric assay (Hitachi 917 analyzer, Boehringer Mannheim, Germany). All samples were stored at −80 °C until analysis, blinded to any clinical information.

### 2.6. Statistical Analysis

Results of normally distributed continuous variables were expressed as the mean ± SD. The normality of distribution was assessed with the Kolmogorov–Smirnov test. Comparisons of continuous and categorical variables were analyzed with the Student’s *t*-test and chi-square test, respectively. Simultaneous comparison of GRT, GCT, and GCO was performed using one-way ANOVA and the post hoc Tuckey test. At baseline, we performed a logistic regression analysis, for normally distributed variables, to assess the univariate and multivariate associations of carotid atherosclerosis presence (patients from both GRT and GCT) with any characteristic of the study population to obtain the odds ratio (OR). We examined Cox regression analysis for the prediction of MACE during follow-up. Hazard ratios (HR) and 95% confidence intervals (CIs) were calculated for each factor. All baseline variables with *p* < 0.05 in univariate analyses were integrated into the Cox multivariate model to determine the independent predictors of clinical end-points. A two-tailed *p* < 0.05 was considered to be significant. The computer software package SPSS (version 28.0; SPSS Inc., Chicago, IL, USA) was used for statistical analysis.

## 3. Results

During follow-up (mean duration: 55 months), eight patients were lost for unknown reasons (three from GRT and five from GCT). In addition, 11 patients died for reasons unrelated to cardiovascular diseases (2 patients due to cancer, 2 patients due to accidents, 7 patients due to infections). Another 18 alive patients were also excluded during follow-up because they fulfilled exclusion criteria: peri-procedural cerebrovascular adverse events (N = 11; TIAs in 8 patients, ischemic strokes in 2 patients, and hemorrhagic stroke in 1 patient) or new onset of excluded conditions (N = 7; cancer, severe renal failure, severe heart failure). A total number of 262 patients from GRT and 109 patients from GCT completed the study and we retrieved full data for analysis.

### 3.1. Baseline Results

The clinical and biochemical characteristics of both groups at baseline are shown in Table 1. Notably, both groups of carotid atherosclerosis had received previously guideline-directed medical therapy (GDMT) and lifestyle advice in the context of secondary prevention. This explains: (1) the higher prescription rate of statins leading to a more favorable lipid profile; (2) the higher prescription rate of anti-hypertensive drugs compensating blood pressure levels; and (3) the lower incidence of active smokers in patients with carotid atherosclerosis compared to controls. Despite GDMT, high levels of CAVI, MMP-3, MMP-7, MMP-9, and TIMP-1 levels were observed across GRT, GCT, and GCO, stratifying cardiovascular risk according to their background (Table 1).

At baseline, we examined in the whole study cohort the correlations of established carotid atherosclerosis (group of revascularization and conservative group) with the other clinical, biochemical, and hemodynamic parameters. In univariate analysis, age, CAD, diabetes, CAVI, serum MMP-7, MMP-9, and TIMP-1 correlated with established carotid atherosclerosis. In logistic regression analysis, CAD (OR = 1.56, *p* < 0.001), diabetes (OR = 1.72, *p* = 0.023), CAVI (OR = 1.29, *p* = 0.012), serum MMP-9 (OR = 1.38, *p* < 0.001) remained independent predictors of significant carotid atherosclerosis (Table 2).

Within the GRT group, we further examined the discriminatory power of variables between asymptomatic vs. symptomatic patients. The symptomatic subgroup showed persistently elevated CAVI, MMP-3, MMP-7, MMP-9, and TIMP-1 levels and lower GSM scores (*p* < 0.001). No significant differences were found in the remaining variables between these subgroups (*p* > 0.05).

### 3.2. Follow-Up Results and Prognosis

During follow-up, we recorded MACE in 56 patients: 51 within GRT and 5 patients within GCT. Among MACE patients, six of them eventually died (five in GRT and one in GCT). We further analyzed the characteristics of patients with MACE compared to event-free counterparts only within GRT, since it was impossible for statistical reasons to analyze markers according to the risk of MACE in five GCT patients. (Table 3). The comparative evaluation revealed a higher prevalence of diabetes and smoking in MACE than the event-free subgroup (*p* < 0.001). Most importantly, we observed a worse risk profile, clustering more indices of atherosclerotic plaque vulnerability, in MACE patients. In particular, they appeared with higher hs-CRP, MMP-7, MMP-9, CAVI, and lower GSM levels compared to event-free patients (*p* < 0.05). No significant differences were detected in the rest of the parameters. 

In Cox regression analysis at baseline, established cardiovascular risk factors, such as diabetes and smoking, along with low GSM score and high CAVI, and MMP-9 emerged as independent predictors of MACE in patients with carotid atherosclerosis requiring revascularization (Table 4). Notably, all patients were already in GDMT. 

Figure 1 summarizes all findings.

## 4. Discussion

In the present study, patients requiring carotid revascularization appeared with higher levels of CAVI, MMP-3, MMP-7 MMP-9, and TIMP-1 compared to patients with carotid atherosclerosis under conservative treatment or controls without carotid atherosclerosis. In the whole study cohort, CAVI, serum MMP-9, CAD, and diabetes emerged as independent predictors of carotid atherosclerosis presence. Most importantly, diabetes and smoking, together with a low GSM score and high CAVI and MMP-9, independently predicted the occurrence of MACE in the post-revascularization period, which has not previously been studied.

MMPs constitute a large family of proteinases, mostly secreted by inflammatory cells within atherosclerotic lesions and degrading extracellular matrix proteins [33,34]. High circulating MMP-3 levels are associated with coronary and carotid atherosclerosis progression [35,36,37]. Plasma MMP-7 concentration might be a marker of atherosclerosis [38]. In contrast to MMP-3 and MMP-7, there are plenty of experimental and clinical studies documenting the pro-atherogenic role of MMP-9 [39,40]. TIMP-1 seems to counteract MMP-9 activity. Elevated circulating TIMP-1 levels in patients with CAD [41], carotid stenosis [42], and lower limb atherosclerosis have long been reported [22], but not in all studies [43]. The elevation of TIMP-1 blood levels may reflect a compensatory mechanism to suppress the pro-inflammatory, pro-atherogenic, and destabilizing effects of MMP-9. One of the most important findings of our study was the graded elevation of MMPs and TIMP-1 along groups divided according to the degree of carotid atherosclerosis. Moreover, symptomatic patients appeared with higher levels in the aforementioned biomarkers than their asymptomatic counterparts. Based on those significant differences between groups (GRT vs. GCT vs. GCO), we hypothesized that both vulnerability and severity of carotid atherosclerosis proportionally influenced the serum levels of those novel biomarkers. Moreover, MMP-3 and MMP-9 have been associated with in-stent restenosis, especially among diabetic patients [44,45]. However, we could not draw cut-off values for risk stratification.

We previously published results about the association between CAVI and the presence of carotid atherosclerosis in a different cohort [46]. In the present study, patients with carotid atherosclerosis, even at a moderate degree, had at least two times higher levels of the examined novel biomarkers. After multivariate analysis, we confirmed CAVI among other well-known independent predictors (CAD, diabetes, serum MMP-9) of carotid atherosclerosis. The discriminatory power of CAVI was small, but statistically significant, to distinguish patients with different degrees of carotid stenosis from patients without carotid atherosclerosis. It would be wise for future studies to test arterial stiffness among patients with atherosclerotic manifestations for stratification purposes. 

Increased expression of both MMP-3 and MMP-7 within human atherosclerotic plaques accompanied by high blood levels have been associated with carotid plaque rupture [47,48]. Similarly, patients with unstable carotid plaques usually show elevated plasma levels of MMP-9 [49] and TIMP-1 [50]. In agreement with previous studies, our symptomatic patients were characterized by more echo-lucent carotid plaques and higher serum levels of MMPs and TIMP-1, supporting a link, causative or bystander, of those factors with carotid plaque vulnerability. Nevertheless, the validation of those biomarkers as potential predictors of carotid plaque instability is pending.

To our knowledge, a limited number of studies have examined the prognostic value of imaging and circulating indices after carotid revascularization [51]. Previous studies were small with heterogeneous populations and focused on peri-operative complications attributed mainly to carotid plaque vulnerability. Concomitant GDMT may indeed diminish the procedures-related complications. However, a significant proportion of patients undergoing uncomplicated carotid revascularization are still exposed to a substantial risk of MACE in the post-procedure long term [52]. Those cardiovascular events and deaths are common among “vulnerable patients” accumulating many traditional and non-traditional cardiovascular risk factors [13,53]. A novelty of our study was to assess the risk profile and MACE occurrence after carotid revascularization. In Cox regression analysis, we identified diabetes, smoking, high CAVI and MMP-9 levels, and low GSM as independent predictors of MACE. High CAVI measurements indicate a systematic mechanism of high cardiovascular risk, despite the absence of a direct relationship with atherosclerosis evolution. MMP-9 degrades stabilizing extracellular proteins and protective fibrous caps due to higher inflammatory activity [54]. In parallel, the low GSM presumably implicates a systematic process of plaque destabilization at other sites (e.g., coronary arteries). Any regimen aiming at carotid plaque stabilization could simultaneously lower the systematic cardiovascular risk, “stabilizing vulnerable patients” [55]. Theoretically, early detection of “vulnerable patients” and prompt intensive therapy could be an effective approach to minimize the risk of MACE. In our cohort, almost all patients with carotid atherosclerosis were already on GDMT. For instance, the prescription rate of statins before enrolment achieved high levels (84–97%). This may explain the lack of any correlation of dyslipidemia with carotid atherosclerosis incidence or MACE. The GDMT prevented us from uncovering all potential cardiovascular risk factors in our multivariate diagnostic and prognostic analysis. Despite GDMT, the remaining significant cardiovascular risk and the associated MACE confirms the hypothesis that the “vulnerable patient” is at risk, and should stimulate further research. 

There are several drawbacks to the present study. The assay of circulating blood levels of novel biomarkers, even along different degrees of carotid atherosclerosis, and their impact on long-term results does not provide definite proof of causality. Notwithstanding the large sample, the cross-sectional design of our study with unadjusted factors, such as co-comorbidities, might have confounded the measurement of circulating levels. On the other hand, all participating patients with carotid atherosclerosis were on GDMT for the whole study period, limiting as much as possible the variable impact of different medications. A second important limitation of the present study was the absence of repeated measurements. Such an approach would have restricted the variation of biomarkers concentrations at baseline, giving a more causative relationship between biomarkers and events. Thirdly, symptomatic patients entered our study at the sub-acute or chronic phase after an event (>15 days). Perhaps we did not assess the excessive inflammatory reaction at the acute phase and our study might have underestimated the pro-inflammatory burden. In addition to this, the application of GDMT might have mitigated the differences in the risk factor pattern between groups. Finally, we excluded patients with significant peripheral artery disease in order to obtain valid CAVI measurements. This might have biased our study by excluding patients with even higher cardiovascular risk.

In conclusion, patients with established carotid atherosclerosis presented with exaggerated CAVI, MMP-3, MMP-7 MMP-9, and TIMP-1 levels compared to individuals without carotid atherosclerotic plaques. Most importantly, the magnitude of elevation paralleled the degree of carotid stenosis and was even greater in symptomatic patients, implicating a high correlation with carotid plaque vulnerability. Regarding the post-intervention follow-up period, the presence of diabetes, smoking, high CAVI and MMP-9 levels, and low GSM independently predicted the occurrence of MACE, despite the GDMT received by all patients. Future studies are required to elaborate on the profile of “vulnerable patients” and the required effective therapy.

## Figures and Tables

**Figure 1 biomolecules-13-01427-f001:**
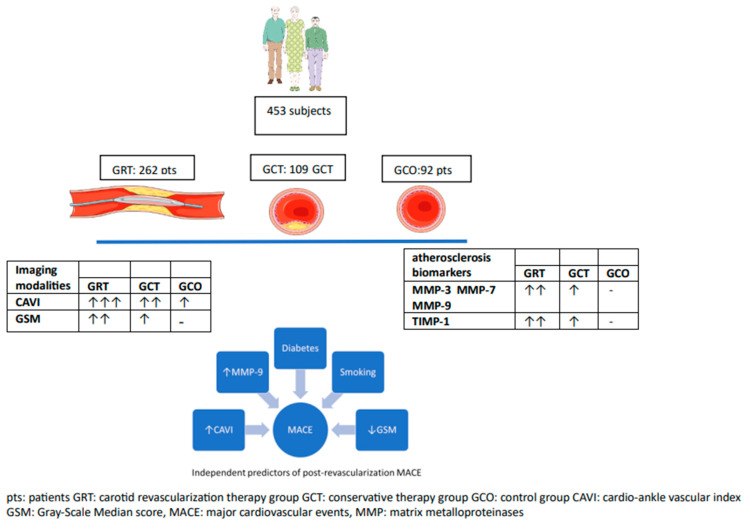
A summary of the most important findings of the study.

**Table 1 biomolecules-13-01427-t001:** Baseline clinical, biochemical, and hemodynamic characteristics of the following groups: carotid revascularization therapy (GRT), conservative therapy (GCT), and controls (GCO).

	GRT (N = 262)	GCT (N = 109)	GCO (N = 92)	*p*
**Age, years**	74 ± 11	70 ± 12	65 ± 13	0.111
**Males/females**	201/61	88/21	70/22	0.775
**CAD, n**	104 (41%)	36 (33%)	0 (0%)	-
**Diabetes, n**	79 (30.1%)	37 (33.9%)	11 (12%)	<0.001
**Smoking, n**	45 (17.2%)	27 (25%)	30 (32.6%) *	0.015
**Statins, n**	254 (97%)	92 (84.4%)	43 (46.7%) *#	<0.001
**Anti-hypertensive medications, n**	231 (88.2%)	92 (84.4%)	57 (62%) *#	<0.001
**BMI, kg/m** ^ **2** ^	29.8 ± 4.2	29.4 ± 3.8	27.5 ± 2.9	0.099
**SBP, mmHg**	128 ± 14	125 ± 12	131 ± 15	0.641
**DBP, mmHg**	80 ± 9	80 ± 10	82 ± 10	0.956
**FPG, mg/dL**	121 ± 23	117 ± 14	94 ± 15 *#	<0.001
**hsCRP (mg/L)**	5.6 ± 1.1	3.5 ± 0.9	1.7 ± 0.6 *#	<0.001
**WBC, cells/μL**	10,051 ± 3114	8234 ± 2995	7580 ± 1816 *	0.031
**TChol, mg/dL**	159 ± 35	151 ± 28	191 ± 39 *#	<0.001
**HDL-C, mg/dL**	41 ± 11	44 ± 10	48 ± 11 *	0.022
**LDL-C, mg/dL**	92 ± 25	85 ± 22	123 ± 22 *#	<0.001
**TG, mg/dL**	128 ± 49	111 ± 49	101 ± 31	0.276
**CAVI, m/s**	10.45 ± 2.08	9.42 ± 1.55	7.58 ± 1.03 *#	<0.001
**GSM score**	65 ± 18	91 ± 22	-	-
**MMP-3 (ng/mL)**	68.03 ± 36.12	49.3 ± 24.23	22.67 ± 6.88 *#	<0.001
**MMP-7 (ng/mL)**	39.77 ± 8.08	30.55 ± 7.49	11.45 ± 2.91 *#	<0.001
**MMP-9 (ng/mL)**	555.59 ± 71.13	375 ± 51.93	185.32 ± 39.14 *#	<0.001
**TIMP-1 (ng/mL)**	118.58 ± 19.09	236.65 ± 31.98	368.47 ± 66.82 *#	<0.001
**MMP-9/TIMP-1 ratio**	4.68 ± 0.82	1.59 ± 0.30	0.5 ± 0.09 *#	<0.001

Data are expressed as mean ± SD or number of participants (n) and (%). *p* value of post hoc Tuckey test. * *p* < 0.05 GCO vs. GRT. # *p* < 0.05 GCO vs. GCT. BMI, body mass index; CAD, coronary artery disease; CAVI, cardio-ankle vascular index; DBP, diastolic blood pressure; FPG, fasting plasma glucose; GSM, Gray-Scale Median; HDL-C, high-density lipoprotein cholesterol; hsCRP, high-sensitivity C-reactive protein; LDL-C, low-density lipoprotein cholesterol; MMPs, matrix metalloproteinases; SBP, systolic blood pressure; TChol, total cholesterol; TG, triglycerides; TIMP-1, tissue inhibitor of matrix metalloproteinase-1; WBC, white blood cells count.

**Table 2 biomolecules-13-01427-t002:** Independent variables associated with carotid atherosclerosis using logistic regression analysis.

	Carotid Atherosclerosis		
Variables	OR	*p*	95% CI
**Age**	1.34	0.591	1.101–1.505
**CAD**	1.56	<0.001	1.331–1.723
**Diabetes**	1.72	0.023	1.371–2.202
**CAVI**	1.29	0.012	1.201–1.429
**MMP-7**	1.09	0.156	−0.01–0.175
**MMP-9**	1.38	<0.001	1.212–1.504
**TIMP-1**	0.87	0.349	0.77–0.99

CAD, coronary artery disease; CAVI, cardio-ankle vascular index; CI, confidence intervals; MMPs, matrix metalloproteinases; OR, odds ratio; TIMP-1, tissue inhibitor of matrix metalloproteinase-1.

**Table 3 biomolecules-13-01427-t003:** Among patients with significant carotid atherosclerosis undergoing revascularization, we compared baseline clinical, biochemical, and hemodynamic characteristics of event-free patients versus patients with MACE.

	Event-Free (N = 211)	MACE (N = 51)	*p*
**Age, years**	76 ± 12	73 ± 12	0.798
**Males/females, n**	167/54	44/7	0.775
**CAD, n**	78 (37%)	26 (51%)	0.081
**Diabetes, n**	57 (27%)	22 (43.1%)	<0.001
**Smoking, n**	25 (11.8%)	20 (39.2%)	<0.001
**Statins, n**	206 (97.6%)	48 (94.2%)	0.923
**Anti-hypertensive medications, n**	185 (87.7%)	46 (90.5%)	0.906
**BMI, kg/m^2^**	29.6 ± 4	30.3 ± 4.3	0.519
**SBP, mmHg**	131 ± 15	125 ± 13	0.763
**DBP, mmHg**	80 ± 8	80 ± 10	0.977
**FPG, mg/dL**	118 ± 21	127 ± 20	0.102
**hsCRP (mg/L)**	5.1 ± 1.2	6.9 ± 1.5	0.025
**WBC, cells/μL**	9807 ± 2456	10,889 ± 3225	0.045
**TChol, mg/dL**	156 ± 37	164 ± 32	0.765
**HDL-C, mg/dL**	43 ± 10	38 ± 9	0.081
**LDL-C, mg/dL**	89 ± 25	100 ± 28	0.228
**TG, mg/dL**	120 ± 48	131 ± 54	0.276
**CAVI, m/s**	10.45 ± 1.08	9.42 ± 1.55	0.010
**GSM score**	65 ± 18	91 ± 22	<0.001
**MMP-3 (ng/mL)**	65.03 ± 36.12	79.55 ± 29.32	0.040
**MMP-7 (ng/mL)**	36.74 ± 8.08	51.01 ± 8.23	<0.001
**MMP-9 (ng/mL)**	490.91 ± 71.13	826.39 ± 91	<0.001
**TIMP-1 (ng/mL)**	128.59 ± 12.11	81.13 ± 5.89	<0.001
**MMP-9/TIMP-1 ratio**	3.82 ± 0.79	10.18 ± 2.1	<0.001

Data are expressed as mean ± SD or number of participants (n) and (%). BMI, body mass index; CAD, coronary artery disease; CAVI, cardio-ankle vascular index; DBP, diastolic blood pressure; FPG, fasting plasma glucose; GSM, Gray-Scale Median; HDL-C, high-density lipoprotein cholesterol; hsCRP, high-sensitivity C-reactive protein; LDL-C, low-density lipoprotein cholesterol; MMPs, matrix metalloproteinases; SBP, systolic blood pressure; TChol, total cholesterol; TG, triglycerides; TIMP-1, tissue inhibitor of matrix metalloproteinase-1; WBC, white blood cells count.

**Table 4 biomolecules-13-01427-t004:** Uni- and multi-variable Cox proportional hazard models of major adverse cardiovascular events within the carotid revascularization group.

Variable	Univariable HR (95% CI)	*p*	Multivariable HR (95% CI)	*p*
**Age (years)**	1.13 (1.01–1.18)	0.031	1.05 (0.98–1.10)	0.451
**Male gender**	1.05 (0.98–1.13)	0.045	1.01 (0.95–1.10)	0.881
**Smoking**	2.28 (1.69–2.92)	<0.001	1.67 (1.35–1.95)	<0.001
**Diabetes mellitus**	2.82 (2.28–3.55)	<0.001	2.07 (1.55–2.78)	<0.001
**GSM**	1.88 (1.29–2.24)	<0.001	1.40 (1.16–2.12)	0.002
**CAVI**	1.40 (1.10–1.95)	<0.001	1.22 (1.09–1.43)	0.023
**MMP-7**	1.22 (1.02–1.88)	0.025	1.11 (1.01–1.54)	0.125
**MMP-9**	1.59 (1.24–2.22)	<0.001	1.44 (1.29–2.15)	0.005
**TIMP-1**	0.72 (0.51–0.97)	0.043	0.92 (0.70–0.99)	0.346
**MMP9/TIMP-1 ratio**	1.72 (1.32–2.59)	<0.001	1.39 (1.11–2.22)	0.055

## Data Availability

Not applicable.

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
