# Peer review of "Novel Biomarkers and Imaging Indices for the “Vulnerable Patient” with Carotid Stenosis: A Single-Center Study"

_biomolecules, 2023, doi:10.3390/biom13091427_

Round 1
Reviewer 1 Report
This is an original research paper about the correlation of various biochemical markers (metalloproteinases etc.) with the degree of carotid artery stenosis in a varied cohort of patients. I particularly liked that the authors included revascularized patients and that they follow-up the patients on the long term. Moreover, the sample size population is relatively large therefore their conclusion is supported by the results.
I have some suggestions for the authors which I kindly ask them to take into account:
1. Abstract. The phrase "During follow-up (mean duration:55 months), 51 25 GRT-patients experienced MACE unrelated to revascularization procedure, while 211 counterparts 26 were event-free. Diabetes, smoking, high CAVI and MMP-9 and low GSM independently predicted 27 MACE occurrence, despite the optimum medical therapy during the study" need actual numbers/ data. Moreover, there is a type there - it's occurrences not occurrence.
2. Please change the reference numbering from Roman numbers (VIII, IX.. etc) to actual numbers. I am not aware that MDPI guidelines require that, it is much easier to follow your references with standard numbers.
3. Introduction. To your phrase "Many researchers have used the term “vulnerable patient” defined as a patient with established atherosclerosis throughout the artery tree, who clusters a number of cardiovascular risk factors, traditional and novel, further precipitating plaque vulnerability and exaggerating the overall cardiovascular risk" please add another reference - https://doi.org/10.1155/2022/5108389. This is an interesting study that actually showed the harmful interplay of atherosclerosis in different arterial districts, and even radial artery atherosclerosis predicts coronary artery atherosclerosis.
4. The next paragraph beginning with "There are many imaging indices of carotid plaque vulnerability" is too long. Make it shorter, 1-2 sentences is enough to say the main idea about CAVI.
5. Results. Clearly written. No major concerns.
6. Discussion. The Discussion section is a bit too long and heavy to digest. Make it friendlier. Begin with a phrase with your key main findings. After that, add some similar studies and see if your data is consistent with that. Add then your personal opinion and interpretation of results, current debates, then some future perspectives, and last, some limitations of your study. That's it.
For example, the phrase "MMPs constitute a large family of proteinases, mostly secreted by inflammatory cells 299 within atherosclerotic lesions [xxxii]. MMP-3, also known as stromelysin, is an MMP mem- 300 ber capable of degrading extracellular matrix major components: collagens I, IV, V, IX and 301 X, proteoglycans, gelatin, fibronectin, laminin, and fibrillin-1 and activating other MMPs 302 during matrix remodeling [xxxiii]. " can be completely deleted.
And so on. Take a look and see if you can lighten it up a bit.
7. Discussion. Add a short discussion about Matrix Metalloproteinases MMP-3 and MMP-9 being also predictors of in-stent restenosis after revascularization. This is important as plaque activation and inflammation negatively affect the physiological endothelization of the carotid stent (here please cite this important study: doi: 10.1016/j.atherosclerosis.2009.05.036). Moreover, diabetes mellitus, like in your study also is a predictor for carotid stent restenosis and failure. This was shown in this recent study - https://doi.org/10.1155/2022/4196195, please cite it as well.
8. Your study has no images. Draw maybe one central abstract image - where you sum up in a visual manner your study's key findings.
No major concerns.
Author Response
This is an original research paper about the correlation of various biochemical markers (metalloproteinases etc.) with the degree of carotid artery stenosis in a varied cohort of patients. I particularly liked that the authors included revascularized patients and that they follow-up the patients on the long term. Moreover, the sample size population is relatively large therefore their conclusion is supported by the results.
I have some suggestions for the authors which I kindly ask them to take into account:
- Abstract. The phrase "During follow-up (mean duration:55 months), 51 25 GRT-patients experienced MACE unrelated to revascularization procedure, while 211 counterparts 26 were event-free. Diabetes, smoking, high CAVI and MMP-9 and low GSM independently predicted 27 MACE occurrence, despite the optimum medical therapy during the study" need actual numbers/ data. Moreover, there is a type there - it's occurrencesnot occurrence.
We really appreciate this important suggestion from the reviewer. We have added the results of Cox regression analysis (The abstract should be a total of about 200 words maximum)
- Please change the reference numbering from Roman numbers (VIII, IX.. etc) to actual numbers. I am not aware that MDPI guidelines require that, it is much easier to follow your references with standard numbers.
Thank you for this notice. It was attributed to endnote transformation. We have corrected that.
- Introduction. To your phrase "Many researchers have used the term “vulnerable patient” defined as a patient with established atherosclerosis throughout the artery tree, who clusters a number of cardiovascular risk factors, traditional and novel, further precipitating plaque vulnerability and exaggerating the overall cardiovascular risk" please add another reference - https://doi.org/10.1155/2022/5108389. This is an interesting study that actually showed the harmful interplay of atherosclerosis in different arterial districts, and even radial artery atherosclerosis predicts coronary artery atherosclerosis.
Lines 54-55: Following reviewers suggestion we have added the reference with the related comment.
- The next paragraph beginning with "There are many imaging indices of carotid plaque vulnerability" is too long. Make it shorter, 1-2 sentences is enough to say the main idea about CAVI.
Lines 58-65: Following reviewer’s suggestion we have shortened the paragraph. However, we left 2 sentences for GSM as well since we have used it in our study and we should give explanations for it.
- Results. Clearly written. No major concerns.
- Discussion. The Discussion section is a bit too long and heavy to digest. Make it friendlier. Begin with a phrase with your key main findings. After that, add some similar studies and see if your data is consistent with that. Add then your personal opinion and interpretation of results, current debates, then some future perspectives, and last, some limitations of your study. That's it.
For example, the phrase "MMPs constitute a large family of proteinases, mostly secreted by inflammatory cells 299 within atherosclerotic lesions [xxxii]. MMP-3, also known as stromelysin, is an MMP mem- 300 ber capable of degrading extracellular matrix major components: collagens I, IV, V, IX and 301 X, proteoglycans, gelatin, fibronectin, laminin, and fibrillin-1 and activating other MMPs 302 during matrix remodeling [xxxiii]. " can be completely deleted.
And so on. Take a look and see if you can lighten it up a bit.
Following reviewer’s suggestion, we have reduced by 25 % the length of the discussion to help the reader to follow it.
- Discussion. Add a short discussion about Matrix Metalloproteinases MMP-3 and MMP-9 being also predictors of in-stent restenosis after revascularization. This is important as plaque activation and inflammation negatively affect the physiological endothelization of the carotid stent (here please cite this important study: doi: 10.1016/j.atherosclerosis.2009.05.036). Moreover, diabetes mellitus, like in your study also is a predictor for carotid stent restenosis and failure. This was shown in this recent study - https://doi.org/10.1155/2022/4196195, please cite it as well.
Lines 306-308: Following reviewer’s suggestion we have added a comment supported by the proposed references.
- Your study has noimages. Draw maybe one central abstract image - where you sum up in a visual manner your study's key findings.
This is a very good suggestion and now we have added a graphical abstract
Reviewer 2 Report
I read with interest the case report by Nikolaos Kadoglou et al. regarding the relationship of CAVI and novel 84 biomarkers (like MMPs and TIMP1) with: 1) the presence and the ultrasonographically 85 quantified severity of established carotid atherosclerosis; 2) the carotid plaque vulnerability; 3) the long-term incidence of MACE after carotid revascularization.
The conclusion of this study highlights that novel imaging and biochemical biomarkers were positively associated with atherosclerosis severity, while CAVI, MMP-9 and low GSM, showed a positive, independent relationship with MACE after carotid revascularization, describing "vulnerable patients".
The article is very well structured and easy to read.
It would be very interesting if the authors analyzed all the markers according to the risk of MACE separately for the GRT and GCT group.
I also want to congratulate the authors for the article and for the work done.
Author Response
I read with interest the case report by Nikolaos Kadoglou et al. regarding the relationship of CAVI and novel 84 biomarkers (like MMPs and TIMP1) with: 1) the presence and the ultrasonographically 85 quantified severity of established carotid atherosclerosis; 2) the carotid plaque vulnerability; 3) the long-term incidence of MACE after carotid revascularization.
The conclusion of this study highlights that novel imaging and biochemical biomarkers were positively associated with atherosclerosis severity, while CAVI, MMP-9 and low GSM, showed a positive, independent relationship with MACE after carotid revascularization, describing "vulnerable patients".
The article is very well structured and easy to read.
It would be very interesting if the authors analyzed all the markers according to the risk of MACE separately for the GRT and GCT group.
I also want to congratulate the authors for the article and for the work done.
We really appreciate the valuable and encouraging comments of the reviewer. In the present analysis we have focused on GRT because 51 patients presented with MACE. In the GCT we had only 5 patients with MACE, so it was underpowered to analyse the risk of MACE in this subgroup. We have added the related comment (Lines 259-260).
Round 2
Reviewer 1 Report
I congratulate Kadoglou et al for their revision. Looking again throughout the manuscript, I believe it presents in a much better and friendlier form.
I have 1 last minor suggestion regarding your title:
- "Novel biomarkers and imaging indices for the "vulnerable patient" with carotid stenosis: a single-center study"
In this way, the reader rapidly understands that this paper is a study not a review, for example.
Other than this, I have no further comments to add. I particularly liked the new Figure 1 that you have inserted.
Sincerely,
The Reviewer.
No Major Issues.
Author Response
Thank you for your comments! We have modified the title as you suggested.
